# GTA Weldability of Rolled High-Entropy Alloys Using Various Filler Metals

**Hyunbin Nam [1], Seonghoon Yoo [1], Junghoon Lee [2], Youngsang Na [3], Nokeun Park [4]**
**and Namhyun Kang [1,\*]**

[1] Department of Materials Science and Engineering, Pusan National University, Busan 46241, Korea;
hbnam12@pusan.ac.kr (H.N.); tjdgns3373@naver.com (S.Y.)

[2] Yonsei University KIURI Institute, Yonsei University, Seoul 03722, Korea; jhoon.lee@yonsei.ac.kr

[3] Titanium Department, Korea Institute of Materials Science, Gyeongnam 51508, Korea; nys1664@kims.re.kr

[4] School of Materials Science and Engineering, Yeungnam University, Gyeongsan 38541, Korea;
nokeun_park@yu.ac.kr

\* Correspondence: nhkang@pusan.ac.kr; Tel.: +82-51-510-3027

**Abstract:** Gas tungsten arc (GTA) weldability of rolled CoCrFeMnNi high-entropy alloys (HEAs) was conducted using stainless steel (STS) 308L and HEA fillers. Microstructure and mechanical properties of the welds were examined to determine GTA weldability of the rolled HEA. The welds had no macro-defects, and component behaviour between base metal (BM) and weld metal (WM) showed significant differences in the weld using the STS 308L filler. Macro-segregation of Fe components was confirmed in the central region in the WM using the STS 308L filler. Because the columnar grain sizes of all the WMs were larger than those of the rolled HEA BM irrespective of the filler metals, the tensile properties of the GTA welds were lower than those of the rolled HEA BM, and the tensile fracture occurred in the centreline of each weld. In particular, the tensile properties of the weld using the STS 308L filler deteriorated more than those of the HEA weld. This was induced by the formation of macro-segregation and severe martensite transformation in the centreline of WM. To enhance the weldability of the rolled HEA, the formation of macro-segregation and coarse grains in the WM of GTA welds must be prevented.

**Keywords:** high-entropy alloys; filler metals; microstructure; macro-segregation; mechanical properties

## 1. Introduction

High-entropy alloys (HEAs) have been highlighted as structural materials for replacing conventional structural steel [1,2]. The high mixing entropy of HEAs stabilises solid solution phases and precludes the formation of embrittling intermetallic compounds [3,4]. Therefore, HEAs are known to have excellent mechanical properties [5,6]. Specifically, CoCrFeMnNi equiatomic HEA, which is a face-centred cubic (FCC)-HEA, has demonstrated excellent cryogenic properties [7–10]. Owing to these advantages, considerable research is ongoing to improve the tensile and cryogenic properties of the base metal (BM) of HEAs [11–14].

To apply HEAs as structural applications, the development of BMs and weldability evaluations should be prioritised [15,16]. HEA BMs have been actively developed; however, studies related to weldability evaluation are limited to low heat input welding: electron beam welding [17], laser beam welding [18–21], friction-stir welding [21–25]. Recently, the evaluation of gas tungsten arc (GTA) weldability on cast HEA using the developed HEA filler has been reported [26], and there have been studies on HEA weldability using high heat input welding [27,28]. The rolled HEA must be welded

for structural applications, because of its finer microstructure and better mechanical properties than those of the cast HEA.

This study investigated GTA weldability using commercial stainless steel (STS) 308L and HEA fillers on rolled CoCrFeMnNi HEAs. Specifically, the microstructural and component behaviours due to the application of various fillers are discussed to understand the tensile properties of the welds.

## 2. Materials and Experimental Procedures

Equiatomic $Co_{0.2}Cr_{0.2}Fe_{0.2}Mn_{0.2}Ni_{0.2}$ was produced using vacuum induction melting. The HEA plates were prepared by homogenising a slab at 1373 K for 24 h, followed by air cooling. The homogenised slab was then hot-rolled from 16 (at 1373 K) to 2 mm (at 1273 K), followed by furnace cooling. The hot-rolled HEA plates were cold-rolled to 1.5 mm at 298 K. Finally, to obtain a plate with a uniform microstructure, cold-rolled HEA plates were annealed at 1073 K for 1 h. Rolled HEA plates had dimensions of 55 mm (W) × 100 mm (L) × 1.5 mm (T).

The HEA BM for welding had a V-groove angle of 30° for feeding the fillers, and the root gap was 0.5 mm. Two types of fillers, HEA ($Co_{0.2}Cr_{0.2}Fe_{0.2}Mn_{0.2}Ni_{0.2}$) and STS 308L (20 wt% Cr, 10 wt% Ni, 1.6 wt% Mn, 0.4 wt% Si, 0.11 wt% (Mo + C), and balanced Fe) were applied to fill the V-groove. The best weldability was obtained from the series of welds prepared under various welding conditions. Finally, GTA welding was performed by a single pass, and the welding conditions were as follows: welding current of 90 A, welding velocity of 10.5 cm/min, wire diameter of Φ 2.1 and Ar shielding gas (purity 99.9%).

The weld pool shape and microstructure of transverse welds were observed using the backscattered electron mode of a scanning electron microscopy (SEM: MIRA3 LMH—TESCAN, Brno, Czech Republic). The microstructure was revealed using the etchant: aqua regia (ethanol 78 mL + hydrochloric acid 18 mL + nitric acid 4 mL) at room temperature for 20 s. To analyse the component behaviour in each region of the welds, the quantitative analysis and macro mapping were performed through electron probe microanalysis (EPMA). In addition, the microstructural behaviour of each region in the welds was confirmed by electron backscattered diffraction (EBSD), with the inverse pole figure (IPF) and image quality (IQ) maps after polishing up to 1 μm and final polishing using colloidal silica. Moreover, the average grain size in the BMs and welds was measured by automatic image analysis of EBSD (ASTM-E2627-13) [29], and the grain size in each region of the welds was averaged from five measurements of IPF images.

Phase analysis of the BM and welds was performed using X-ray diffraction (XRD: ULTIMA4—Rigaku Corp., Tokyo, Japan) and EBSD with a phase map. XRD analysis was performed at a scan speed of 2°/min, with a range of 20–90°, voltage of 40 kV, and current of 30 mA, using Cu $K_\alpha$ radiation. EBSD analysis was conducted at a working distance of 17 mm, with a step size of 0.3 μm. To prove the tensile mechanism of each weld, the fraction of the deformation twins and martensite transformation were measured by the coincidence site lattice (CSL) boundary and phase map of EBSD, respectively.

To observe the hardness distribution of each region in the GTA welds, Vickers hardness measurements in the welds were performed with a load of 300 gf (4.903 N) and dwell time of 10 s. The hardness was measured at a position 0.6 mm below the weld surface, at intervals of 0.5 mm. After flattening the upper and lower beads of the welds, tensile tests were performed on a sub-size based on ASTM E8 [30]. Tensile tests were carried out at 298 K and a strain rate of $8.3 \times 10^{-4}$ s$^{-1}$.

## 3. Results and Discussion

### 3.1. GTA Weldability of Rolled HEA Using Various Fillers

Figure 1 shows the shape of the GTA weld cross-sections produced by the HEA and STS 308L fillers. Sound welds were obtained without macro-defects such as internal pores and cracks. In addition, in all welds, the upper beads were almost flat, and the lower beads were found to have a convex

shape. The fusion lines (blue dotted lines) of the welds were clearly confirmed, irrespective of the fillers. These results show that when each filler was melted, a part of the BM was properly melted, resulting in sound welds irrespective of the fillers.

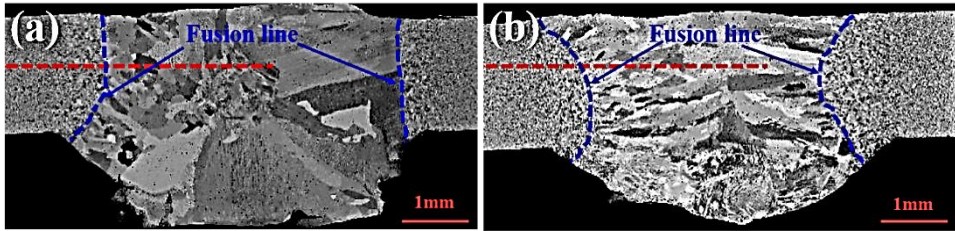

**Figure 1.** Shape of the gas tungsten arc (GTA) weld cross-sections using various filler metals: (**a**) stainless steel (STS) 308L and (**b**) high-entropy alloys (HEA). Red- and blue-dotted lines indicate the location of the quantitative electron probe microanalysis (EPMA) and fusion lines, respectively.

### 3.2. Compositional and Microstructural Behaviour of GTA Welds Using Various Filler Metals

Figure 2 shows the compositional behaviour of the obtained welds. Macro mapping of EPMA showed the compositional segregation behaviour of Fe, Ni, and Cr, the main components of the STS 308L filler (Figure 2a). However, the weld obtained using the HEA filler had a uniform composition with the BM, as shown by the little colour difference (Figure 2b) in the main components between BM and weld metal (WM). Significant variations were observed in the component behaviour of Fe and Ni in each WMs. In the centreline of the weld obtained using STS 380L filler, macro-segregation region of the Fe component and deficient region of the Ni component were formed because of the difference in the alloy compositions of the rolled HEA BM and that of each filler (HEA and STS 308L). The melting temperature of the STS 308 filler (~1723 K) was higher than that of the HEA BM (~1550 K), which is one of the causes of the inhomogeneous mixing during GTA welding [31,32].

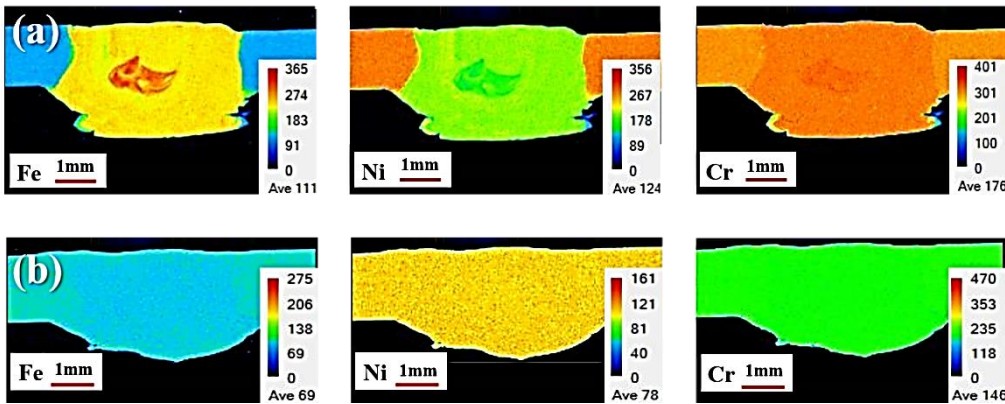

**Figure 2.** Component segregation of GTA welds produced using various fillers: (**a**) STS 308L and (**b**) HEA.

Figure 3 shows the component behaviour quantitatively in the welds using HEA and STS 308L fillers. As depicted in Figure 1, EPMA quantitative analysis was performed along the red dotted lines from the rolled HEA BM to the centreline of the welds produced by the HEA and STS 308L fillers. For the WM produced by STS 308L filler, the composition of Fe in the WM (46.4 ± 0.5 wt.%) was significantly larger than that of the HEA BM (21.4 ± 0.2 wt.%), and slightly lower than that of STS 308L filler (67.9 ± 0.1 wt.%) (Figure 3a). Moreover, the compositions of Co, Ni, and Mn in the WM were lower than those in the HEA BM. The Cr component of the WM exhibited behaviour that was homogeneous with that of the HEA BM. Furthermore, small amounts of Si and C were detected in the WM, as shown in Figure 3a. The WM using STS 308L filler contained higher amounts of Ni, Mn,

and Co than in case of the STS 380L filler, which stabilised the FCC because the STS 380L filler was diluted from the HEA BM to the WM. However, the compositions of all components in the WM using the HEA filler indicated behaviour that was homogeneous with that of the BM (Figure 3b).

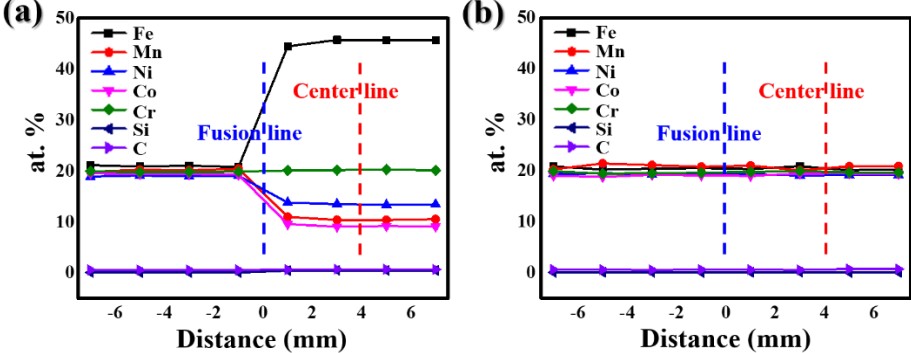

**Figure 3.** Compositional behaviour of GTA welds produced using various fillers: (**a**) STS 308L and (**b**) HEA.

To link compositional behaviour with the weld microstructure, we performed a phase analysis using XRD. Figure 4 shows the crystal structure obtained by XRD in the rolled HEA BM and welds produced using HEA and STS 308L fillers. Irrespective of the filler (HEA and STS 308L), the observed FCC single phase with diffraction peaks ($2\theta$ = 43.4°, 51.6°, and 74.7°) in the welds was the same as that in the rolled HEA BM. Typically, the STS 304 weld using STS 308L filler contains a small amount of δ-ferrite, which prevents solidification cracking during welding [33–35]. However, it was confirmed that the HEA weld using STS 308L filler comprised only the FCC single phase. The body-centred cubic (BCC) phase was not formed in the WM using STS 308L filler, because the stabilising elements of FCC were diluted from the HEA BM [26]. XRD results of the rolled HEA BM and each weld were consistent with those of EPMA quantitative analysis. The WM produced by STS 308L filler contained a higher amount of FCC-stabilising components (Ni, Mn, and Co) than in the STS 308L filler (Figure 3a).

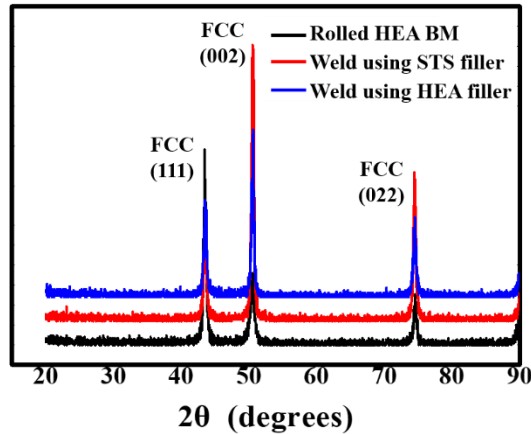

**Figure 4.** XRD patterns of rolled HEA base metal (BM) and GTA welds using HEA and STS 308L fillers.

Figure 5 shows the microstructural behaviour of each region, such as BM, heat-affected zone (HAZ), fusion line, and WM, in the welds using the HEA and STS 308L fillers. The rolled HEA BM had fine equiaxed grains with an approximate size of 6 ± 0.3 μm, and twins generated by recrystallisation of the annealing treatment were observed (Figure 5a). For all welds, the grain morphology of the HAZ was similar to that of the BM. However, the grain size of the HAZ was approximately 9 ± 0.5 μm, which was coarser than that of the BM (Figure 5b). However, when approaching the fusion line, finer

grains were observed. These areas have been reported as partially melted zones (PMZs) in the case of nonferrous metals, such as aluminium and magnesium alloys [36,37].

**Figure 5.** Microstructural behaviour of (**a**) rolled HEA BM, (**b**) HAZ, (**c**–**f**) near the fusion line and WM centreline using various fillers: (**c**, **e**) STS 308L and (**d**, **f**) HEA.

Irrespective of the fine grains generated in the PMZs using STS 308L and HEA fillers, coarse columnar grains occurred from the fusion line and grew in the direction of the weld centreline (Figure 5c,d). The columnar grains in the WM using the HEA filler showed the same colour as the grains in the PMZ, indicating epitaxial growth (Figure 5d). However, columnar grains in the WM produced by STS 308L filler were depicted in different colours than the grains in the PMZ (Figure 5c), which is associated with the significant compositional differences between the rolled HEA BM and WM using STS 308L filler (Figures 1a, 2a and 5c).

Finally, towards the centreline of the WM, coarse columnar grains near the fusion line were transformed into equiaxed grains of the top and middle parts in all welds. However, in the bottom part of the welds, coarse columnar grains were formed in the heat flow direction of the root gap of the V-groove (Figure 5e,f). Equiaxed grains at the centreline of the WM are typical microstructures formed by high heat input welding such as GTA welding [27,31]. Irrespective of the STS 308L and HEA fillers, the total grain size of columnar grains and equiaxed grains in the WM was similar (approximately 400 μm for WM using STS 308L filler and 410 μm for WM using HEA filler).

### 3.3. Mechanical Properties of GTA Welds Using Various Filler Metals

#### 3.3.1. Hardness Distribution Behaviour of the Welds

Figure 6 shows the hardness distribution in the transverse welds produced using the HEA and STS 308L fillers. The average hardness of the rolled HEA BM and WMs was 177 ± 1 Hv and 152 ± 2 $Hv_{0.3}$, respectively. All WMs showed lower average hardness than that of the rolled HEA BM. As shown in Figures 4 and 5, no phase transformation and secondary phase occurred, owing to the welding heat input. The main reason for the significant difference in the average hardness between the HEA BM and WM in all welds was that the grain size of the BM (~6 μm) was approximately 70 times smaller than that of the WM (400–410 μm) in all welds. The largest hardness in the HAZ near the fusion line was associated with the fine grains locally generated in the PMZ, as shown in Figure 5c,d. Therefore, the hardness distribution in each weld was closely related to the grain size of each region (BM/HAZ/WM) in the welds, and the hardness distribution behaviour of each weld was mostly the same, irrespective of the filler.

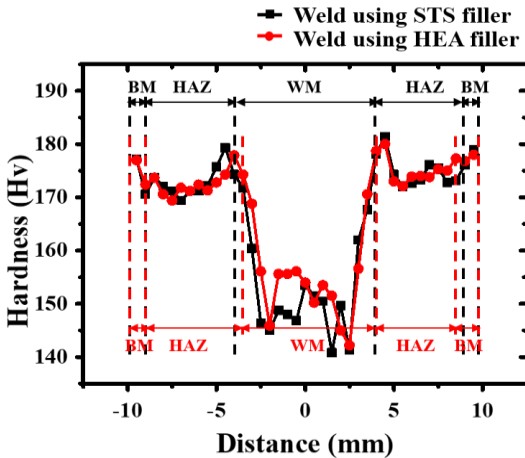

**Figure 6.** Hardness distribution behaviour of the welds using HEA and STS 308L fillers.

### 3.3.2. Tensile and Microstructural Behaviour of the Welds

Figure 7a,b show the tensile stress-strain curves and fracture positions of rolled HEA BM and welds using HEA and STS 308L fillers, respectively, at 298 K. The yield strength (YS), tensile strength (TS), and elongation-to-fracture ($E_f$) of the rolled HEA BM were measured to be approximately 377 MPa, 672 MPa, and 53%, respectively. For welds produced using various fillers (HEA and STS 308L), the tensile properties of all welds were worse than those of the rolled HEA BM. The YS of each weld was approximately 374 ± 2 MPa, which was almost the same as that of the BM, and the TS and $E_f$ of the welds using the HEA filler were 58 MPa and 5% higher than those of the welds using the STS 308L filler. Nevertheless, the tensile fracture of all welds occurred near the centreline in the WMs (Figure 7b).

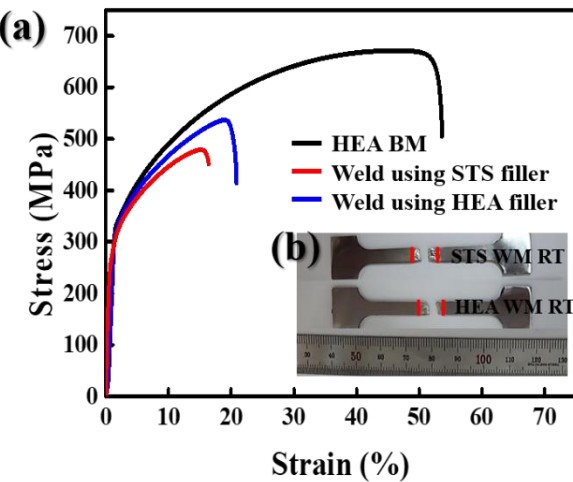

**Figure 7.** Tensile properties of rolled HEA BM and welds using HEA and STS 308L fillers tested at 298 K: (**a**) tensile stress-strain curves and (**b**) fracture positions in the welds.

To understand the mechanism of the tensile properties in the welds, microstructural analysis was performed near the tensile fracture of welds produced using the HEA and STS 308L fillers. As shown in Figure 7b, all welding specimens were fractured near the centreline of the WM. Figure 8 shows the microstructure near the fracture position in the welds using HEA and STS 308L fillers. We analysed the microstructure of each weld via the IPF, IQ, and phase maps of the EBSD. Deformation twins were observed by the CSL boundary of $\Sigma3$, which is indicated by the red line in the IQ maps. The fraction of deformation twins (CSL boundary) of WM using STS 308L filler was 0.08 and that of WM using HEA filler was 0.22. The STS 308L filler produced fewer CSL boundaries than in the case of the HEA

filler. The phase map distinguished austenite (yellow), $\alpha'$-martensite (red), and $\varepsilon$-martensite (green). The STS 308L filler produced more $\alpha'$ and $\varepsilon$ martensites (Ms ~0.24) than the HEA filler (Ms ~0.15). The tendency of fraction of martensite transformation was opposite to that of the CSL boundary, that is, a smaller CSL boundary and more martensite for the weld produced using the STS 308L filler. These results were associated with the macro-segregation observed near the WM centreline (Figure 2a). The FCC stabilising elements of Ni, Mn, and Co were deficient in the WM centreline (Figure 3a), which facilitated the formation of martensite transformation in the WM using the STS 308L filler.

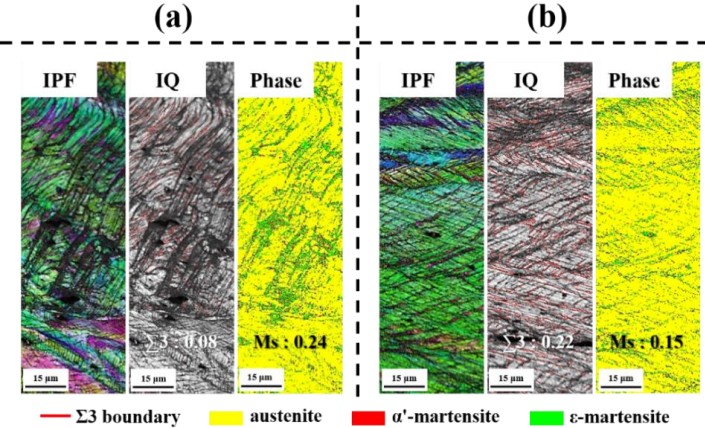

**Figure 8.** Microstructures near the tensile fracture position tested at 289 K in each weld: (**a**) using STS 308L filler and (**b**) HEA filler.

Figure 9 shows the fracture morphology of the welds obtained using the HEA and STS 308L fillers tested at 298 K. The fracture surface of all welds primarily comprised dimple morphology. However, the fracture morphology of quasi-cleavage (QC) was observed more frequently near the central part of the fracture surface of the weld using the STS 308L filler than that using the HEA filler. The large fraction of QC fracture and the deterioration of TS and $E_f$ were associated with considerable macro-segregation and martensite transformation produced in the WM using STS 308L filler. Conclusively, the commercialised STS 308L filler was not sufficient to produce CoCrFeMnNi HEA welds that are stronger than the rolled HEA BM. The authors previously reported the successful use of the STS 308L filler for the cast HEA weld and produced a stronger WM than the BM [26]. Future studies are required to enhance grain refinement and dispersion strengthening to enable the use of the HEA for structural applications.

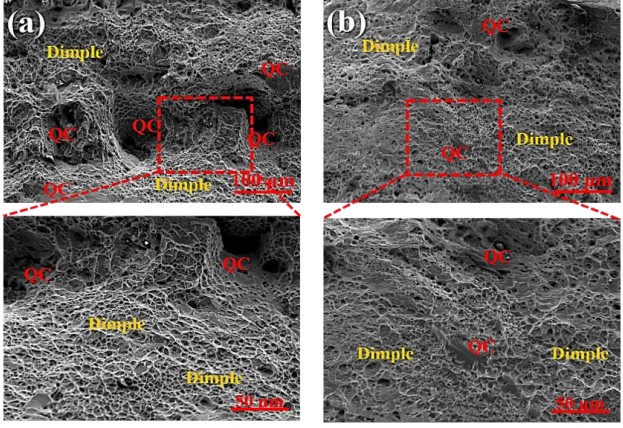

**Figure 9.** Microstructures near the tensile fracture position tested at 289 K in each weld using: (**a**) STS 308L filler and (**b**) HEA filler.

## 4. Conclusions

GTA weldability of the rolled CoCrFeMnNi HEA using STS 308L and HEA fillers was investigated. Microstructures and mechanical properties of the GTA welds were examined at room temperature, and the conclusions of this study are as follows:

(1) No macro-defects such as internal pores or cracks were observed for any of the GTA welds. However, the macro-segregation region of the Fe component and deficient regions of Ni, Mn, and Co components were formed in the centreline of the weld using an STS 308L filler.

(2) For WMs produced using different fillers (HEA and STS 308L), an FCC solid solution phase was observed. The weld using the STS 308L filler had no BCC phase because of the dilution of the stabilising element of FCC introduced from the HEA BM. Furthermore, the columnar grains exhibited unidirectional growth from the fusion line in the WM using the STS 308L filler than those using the HEA filler.

(3) The main reason for the low hardness of the WM was that the grain size of WM was approximately 70 times larger than that of the rolled HEA BM regardless of the filler metals.

(4) The tensile properties of all welds were worse than those of the rolled HEA BM, and the tensile fracture of all welds occurred near the centreline in the WMs. Furthermore, the tensile properties of the weld using the STS 308L filler deteriorated more than those of the weld using the HEA filler. This was associated with the macro-segregation and severe martensite transformation formed in the centreline of WM. Therefore, to enhance the weldability of the rolled HEA, it is necessary to prevent the formation of macro-segregation and enhance grain refinement in the WM of GTA welds.

**Author Contributions:** Conceptualization, writing—review and editing, project administration, funding acquisition, supervision N.K.; writing—original draft preparation, investigation, formal analysis, data curation, H.N.; investigation, formal analysis, S.Y. and J.L.; investigation, resources, N.P. and Y.N., All authors have read and agreed to the published version of the manuscript.

**Funding:** This work was support by the Future Material Discovery Project of the National Research Foundation of Korea (NRF) funded by the Ministry of Science, ICT and Future Planning (MSIP) of Korea (2016M3D1A1023534), and by the World Class 300 Project R&D (S2482209) of the MOTIE and MSS (Korea).

**Acknowledgments:** We would like to thank the personnel responsible of the ESAB SeAH Corp. for providing the welding filler metals and helping to perform the welding for this work.

**Conflicts of Interest:** The authors declare no conflict of interest.

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
