# Peer review of "GTA Weldability of Rolled High-Entropy Alloys Using Various Filler Metals"

_metals, doi:10.3390/met10101371_

Round 1
Reviewer 1 Report
I found the paper by Hyunbin Nam et al. interesting, presenting novel results important for the scientific community.
However, some experimental details are missing in the paper. I would recommend to publish the paper after minor issues listed below are addressed:
- Line 47: authors should provide details about reagents (purity, producer)
- Lines 48-50: it is not clear at what temperature the hot rolling process was done.
- line 58: what grade of Ar gas was used? (technical?)
- line 67: provide details about the XRD setup (lamp material, geometry, equipment used) and preparation of the sample.
Author Response
1. Authors should provide details about reagents (purity, producer).
(Response) Following the reviewers’ comment, the authors added the etchant to observe the macro- and microstructure in the experimental procedures. The authors highlighted the revised texts in the manuscript (Line 60 of page 2) as follows:
“The weld pool shape and microstructure of transverse welds were observed using the backscattered electron mode of a scanning electron microscopy (SEM). The microstructure was revealed using the etchant: aqua regia (ethanol 78 mL + hydrochloric acid 18 mL + nitric acid 4 mL) at room temperature for 20 s.
2. It is not clear at what temperature the hot rolling process was done.
(Response) Following the reviewers’ comment, the authors added the temperature of hot rolling process in the experimental procedures. The authors highlighted the revised texts in the manuscript (Line 49 of page 2) as follows:
“The homogenised slab was then hot rolled from 16 (at 1373 K) to 2 mm (at 1273 K), followed by furnace cooling.”
3. What grade of Ar gas was used? (technical?)
(Response) Following the reviewers’ comment, the authors added the purity grade of Ar gas in the experimental procedures. The authors highlighted the revised texts in the manuscript (Line 58 of page 2) as follows:
“welding current of 90 A, welding velocity of 10.5 cm/min, wire diameter of Φ 2.1 and Ar shielding gas (purity 99.9 %).”
4. Provide details about the XRD setup (lamp material, geometry, equipment used) and preparation of the sample.
(Response) Following the reviewers’ comment, the authors added the detailed information of XRD setup in the experimental procedures. The authors highlighted the revised texts in the manuscript (Line 69 of page 2) as follows:
“Phase analysis of the BM and welds was performed using X-ray diffraction (XRD: ULTIMA4 – Rigaku Corp.) and EBSD with a phase map. XRD analysis was performed at a scan speed of 2 °/min, with a range of 20–90 °, voltage of 40 kV, and current of 30 mA using Cu Kα radiation. EBSD analysis was conducted at a working distance of 17 mm, with a step size of 0.3 μm.”

Reviewer 2 Report
The authors investigated the gas tungsten arc weldability of rolled high-entropy CoCrFeMnNi alloy. Microstructure and mechanical properties of the welds with 308L and HEA filler were analyzed. The manuscript is very similar to other from the same authors [26], however, the publication of the work in Metals may have value after the authors consider the following major points.
- Did the experimental steps comply with some Standard/Specification/published procedure? If yes it needs to be properly referenced.
- Is there any difference in the alloy composition of the rolled HEA and that of HEA filler (line 96-97)?
- Check the reference at line 99.
- Line 108: “the composition of Fe in the WM was significantly larger than that in the HEA BM…”. The weight composition (wt%) of HEA with standard error should be provided.
- What are the experimental conditions and the chemical reagent for the microstructural behavior presented in Fig.5 ? The procedure must be properly explained.
- The authors should provide some results from other techniques (for ej. corrosion test) in order to differentiate the quality of welds or to compare with another high-entropy alloy.
- Must to avoid references like 17-25 … this is meaning superficiality!
- Conclusion nº3 (line 246) must be revised.
Author Response
1. Did the experimental steps comply with some Standard/Specification/published procedure? If yes it needs to be properly referenced.
(Response) Following the reviewers’ comment, the authors added the Standard of each experimental step (welding condition, grain size measurement, tensile test method) in the experimental procedures. The authors highlighted the revised texts in the manuscript (Line 57, 66, 76 of page 2) as follows:
“The average grain size in the BMs and welds were measured by automatic image analysis of EBSD (ASTM-E2627-13) [29], and the grain size in each region of the welds were averaged from five measurements of IPF images.”
“After flattening the upper and lower beads of the welds, tensile tests were performed on a sub-size based on ASTM E8 [30].”
2. Is there any difference in the alloy composition of the rolled HEA and that of HEA filler (line 96-97)?
(Response) The authors mentioned the composition the BM and filler in the manuscript (Line 98 of page 3). The rolled HEA and HEA filler had the same composition as Co20Cr20Fe20Mn20Ni20 (at.%).
3. Check the reference at line 99.
(Response) Following the reviewers’ comment, the authors checked and modified the reference of line 99. The authors highlighted the revised texts in the manuscript (Line 105 of page 3) as follows:
“which is one of the causes of the inhomogeneous mixing during GTA welding [29,30].”
4. Line 108: “the composition of Fe in the WM was significantly larger than that in the HEA BM…”. The weight composition (wt.%) of HEA with standard error should be provided.
(Response) Following the reviewers’ comment, the authors added the Fe composition (wt.%) of WM using STS 308L filler and HEA BM with standard error. The authors highlighted the revised texts in the manuscript (Line 114 of page 3) as follows:
“the composition of Fe in the WM (46.4 ± 0.5 wt.%) was significantly larger than that of the HEA BM (21.4 ± 0.2 wt.%) and slightly lower than that of STS 308L filler (67.9 ± 0.1 wt.%) (Figure 3a).”
5. What are the experimental condition and the chemical reagent for the microstructural behavior presented in Fig. 5? The procedure must be properly explained.
(Response) Following the reviewers’ comment, the authors added the experimental condition and the chemical reagent for the microstructural behavior in the experimental procedures. The authors highlighted the revised texts in the manuscript (Line 65 of page 2) as follows:
“In addition, the microstructural behaviour of each region in the welds was confirmed by electron backscattered diffraction (EBSD) with the inverse pole figure (IPF) and image quality (IQ) maps after polishing up to 1 μm and final polishing using colloidal silica.”
6. The authors should provide some results from other techniques (for ej. Corrosion test) in order to differentiate the quality of welds or to compare with another high-entropy alloy.
(Response) The rolled HEA must be welded for structural applications because of its finer microstructure and better mechanical properties than those of the cast HEA. The authors mentioned the originality of this manuscript as compared with other HEA welds (Line 41 of page 1).
7. Must to avoid references like 17-25… this is meaning superficiality!
(Response) Following the reviewers’ comment, the authors revised the manuscript as follows:
“HEA BMs have been actively developed; however, studies related to weldability evaluation are limited to low heat input welding: electron beam welding [17], laser beam welding [18-21], friction-stir welding [21-25].”
8. Conclusion 3) (line 246) must be revised.
(Response) Following the reviewers’ comment, the authors revised the conclusion 3. The authors highlighted the revised texts in the manuscript (Line 249 of page 8) as follows:
“The main reason for the low hardness of the WM WAS that the grain size of WM was approximately 70 times larger than that of the rolled HEA BM regardless of the filler metals.”

Round 2
Reviewer 2 Report
The authors corrected the manuscript following the recommendations.